# Risk Factors for Female Breast Cancer: A Population Cohort Study

**DOI:** 10.3390/cancers14030788

**Published:** 2022-02-03

**Authors:** Yu-Chiao Wang, Ching-Hung Lin, Shih-Pei Huang, Mingchih Chen, Tian-Shyug Lee

**Affiliations:** 1Graduate Institute of Business Administration, College of Management, Fu Jen Catholic University, New Taipei City 242062, Taiwan; 406088139@mail.fju.edu.tw (Y.-C.W.); 081438@mail.fju.edu.tw (M.C.); 2Department of Medical Oncology, National Taiwan University Cancer Center Hospital, Taipei 10016, Taiwan; chinghlin@ntu.edu.tw; 3Department of Oncology, National Taiwan University Hospital, Taipei 10016, Taiwan; 4Adjunct Department and Graduate Institute of Medical Education and Bioethics, College of Medicine, National Taiwan University, Taipei 10051, Taiwan; huangsp@ntu.edu.tw; 5Intelligent Medical Big Data Co., Ltd., Taipei 10044, Taiwan; 6Artificial Intelligence Development Center, Fu Jen Catholic University, New Taipei City 242062, Taiwan

**Keywords:** female breast cancer incidence, population-based cohort study, sedentary behavior, sugary drink intake

## Abstract

**Simple Summary:**

In recent years, it has been emphasized that the development of a healthy lifestyle can prevent the incidence of cancer, and several recent studies on female breast cancer (BC) have also become interested in sedentary behavioral issues. Our prospective cohort study found that, in addition to the currently known risk factors (RFs) such as parity and body mass index (BMI), which affect the probability of developing BC in women, a sedentary lifestyle and drinking sugar-sweetened beverages (SSB) can cause BC. Therefore, we propose that the modifiable risk profiles of sedentary behavior and sugary beverage consumption may also be associated with increased incidence of female BC in the Taiwanese population.

**Abstract:**

Background: The incidence of female BC among the Eastern and Southeastern Asian populations has gradually increased in recent years. However, epidemiological studies on the relationship between a sedentary lifestyle and female BC are insufficient. In order to determine the association between this lifestyle and the incidence of female BC, we conducted a population-based cohort study on women in Taiwan. Methods: We followed a prospective cohort of 5879 women aged 30 years and over enrolled in the 2001 National Health Interview Survey (NHIS), who developed female BC over a period of 72,453 person years, and we estimated the hazard ratios (HRs) and corresponding 95% confidence intervals (CIs) using the Cox proportional hazards model. Results: RFs associated with female BC incidence included parity (adjusted HR = 0.63; 95% CI: 0.44–0.91), body mass index (adjusted HR = 1.34; 95% CI: 1.04–1.71), and ≥3 h/day spent sitting (adjusted HR = 1.89; 95% CI: 1.08–3.32). The incidence of female BC in participants who sat for ≥3 h/day and consumed sugary drinks was 2.5 times greater than that in those who sat for <3 h/day and did not consume sugary drinks (adjusted HR = 2.51; 95% CI: 1.01–6.23). Conclusions: The findings of this study indicate that sedentary behavior and sugary drink intake may increase the risk of developing female BC. These are modifiable RFs; therefore, a healthy lifestyle and diet can reduce the incidence of female BC.

## 1. Introduction

As of 2018, female BC was the second most frequent type of cancer worldwide and was one of the major threats to life and causes of death in women [1]. Nevertheless, the incidence of BC in women varies greatly between different regions around the world. Generally, the incidence is highest in developed countries, but its incidence has plateaued and has even fallen since 2000. However, female BC is becoming more common in low-incidence regions of Asia [2,3]. Cancer negatively affects the quality of life of patients and their families. However, the recent rapid increase in the incidence of female BC has posed a major new public health challenge. It has even affected the entire healthcare system and has led to an increased economic burden. The current public health response to female BC is to reduce risks, such as improving modifiable RFs to prevent female BC [3].

There are many modifiable RFs that contribute to female BC [4]. One study estimated the population-attributable risk percentage for the risk of female BC, including obesity (10%), a lack of physical activity (3%), a high alcohol intake (3%), oral contraceptive use (3%), hormone replacement therapy (HRT) (7%), and delayed first birth (1%). All of these reproductive factors and lifestyles are modifiable [5]. In addition, previous studies have suggested that age is the most primary RF for female BC. Female BC incidence increases with age, doubling approximately every decade before menopause but dropping rapidly after menopause. However, the curve in some countries tends to be flat [4,6]. As the parity increases, women’s risk of BC decreases significantly (trend *p* = 0.01) [7]. The older the age of menopause onset, the greater the risk of Japanese women developing BC (trend *p* = 0.02) [8]. Those who recently received HRT are more likely to develop female BC (adjusted relative risk (RR) = 1.66; 95% CI: 1.58–1.75) [9]. Compared with no alcohol consumption, each 10 g/day increase in alcohol intake increased the HR for female BC by 4.2% (95% CI: 2.7–5.8) [10]. Smokers have an increased risk of 16% (HR = 1.16; 95% CI: 1.00–1.34) [11]. Each additional unit of saturated fat consumption increases the risk by 20% (HR = 1.2; 95% CI: 1.20–1.21) [12]. Higher BMI positively correlates with risk of female BC (adjusted HR = 0.78, 1.19, 1.31, 1.53, and 1.65; 95% CI: 0.63–0.96, 1.12–1.27, 1.21–1.41, 1.38–1.71, and 1.27–2.13 for BMIs of 18.5, 24–26.9, 27–29.9, 30–34.9 and ≥35 kg/m^2^, respectively) [13]. A higher total current physical activity level negatively correlates with risk of female BC (trend *p* = 0.03) [14].

Recently, a new scope of epidemiology studies has focused on the association between a sedentary lifestyle and cancer. Sedentary behavior means prolonged sitting or reclining, which reduces energy expenditure. It is different from the lack of health-enhancing physical activity. One may meet the recommendations of recreational physical activity but still spend a long time sitting. A systematic review suggests that a sedentary lifestyle may increase cancer risk through factors such as adiposity and metabolic dysfunction [15]. A nested case–control study was conducted to examine the difference between the causes of female BC in Asia and the known RFs in Western countries. The results showed that prior history of breast disease (OR = 2.47; 95% CI: 2.26–2.71), obesity (OR = 1.43; 95% CI: 1.04–1.96), hypertension (OR = 1.14; 95% CI: 1.05–1.25), and some estrogen-related factors may increase the risk of female BC [16]. Thus, our aim was to conduct a population-based cohort study to explore the association between lifestyle and rapid increase in female BC.

## 2. Materials and Methods

### 2.1. Study Cohort and Baseline Characteristics

This study involved female participants from the previous 2001 NHIS cohort [17]. The NHIS is a national cross-sectional study that uses a multi-stage stratified sampling method to collect a sample representative of the Taiwanese population. Interviewers were trained in standardized interview procedures and then conducted face-to-face questionnaire interviews [18]. Briefly, a total of 5911 women (see Figure 1) aged 30 years and over who agreed to link to the health insurance database were included in this study. Those who had a catastrophic illness card (NHIRD’s Registry for Catastrophic Illness Database: ICD, 9th Revision, code 174) prior to the study were excluded. Thus, after exclusion, a total of 5879 women aged 30 years and above were included in this study.

In this study, we focused on reproductive factors, lifestyle, and obesity that are potentially associated with female BC. Information on demographic variables, reproductive factors (parity or number of childbirths and hormone therapy), lifestyle (alcohol intake, cigarette smoking, exercise, time spent sitting, sugary drink intake, meat intake, fruit intake, and vegetable intake), and disease history were derived from a structured questionnaire interview. We converted height and weight into BMI. Those aged over 55 years were considered postmenopausal [19,20]. Exercise refers to any form of exercise performed in the past two weeks. Sedentary time is the average number of hours per day that the participant answered for sitting (besides sleeping), including working, reading, watching TV, and using the computer. Dietary pattern is the consumption frequency of the listed foods in a week: Almost every day, 3–5 times per week, 1–2 times per week, <1 time per week, and none.

### 2.2. Follow-Up and Determination of Female Breast Cancer

We followed the cancer and survival status of the study subjects by using data to link health profiles in the NHIRD’s Registry for Catastrophic Illness and National Cause of Death Registry (NCDR) in Taiwan. The National Health Insurance (NHI) program has been established in Taiwan since 1995. It has nationally insured rates of more than 99% (by 2017, 23.88 million people were insured) [21,22,23]. The National Health Insurance Research Database (NHIRD) provides, in electronic format, all medical benefit data of NHI beneficiaries during the insurance cancelation period. The NHIRD’s Registry for Catastrophic Illness includes a list of 30 types of illnesses that require long-term care, including cancer. A panel of specific disease experts reviews the original charts, images, and pathology reports of patients who have applied for the catastrophic illness certificate, and the panel verifies the diagnosis [24,25]. In Taiwan, all death certificates issued by doctors must be input into the NCDR database. The cause of death data only provide the main cause of death and the date of death of all the citizens recorded in the NCDR. Previous studies have indicated that the overall agreement rate of the determination of death for malignant neoplasms between reviewers and coders is more than 90% (kappa = 0.94) [26].

All eligible subjects were followed from the enrollment date selected for this study until the onset of cancer, death of any cause, or the end of the follow-up date (31 December 2014), whichever was the earliest. Individuals until the defined follow-up endpoint with a history or subsequent diagnosis of any cancer except female BC were not excluded. The cancer occurrence in our analysis was defined as the incidence of female BC (ICD, 9th Revision, code 174) as a primary diagnosis.

### 2.3. Statistical Analyses

We calculated the follow-up time for each individual from the date of enrollment until the earliest date of newly diagnosed female BC, death, or last follow-up. From using the life table method, we derived the cumulative incidence of female BC for each year of follow-up among subjects based on time spent sitting (T) and sugary drink intake (S) at the time of enrollment. We used the Cox proportional hazard model to identify the relative risk and its 95% CI between RFs and female BC, while presenting the visualization results as the forest plot in Table 1. In order to ensure tightly controlled potential confounding effects, we controlled for age as a continuous variable in the multivariate Cox regression analysis. A two-tailed test was used for all Cox regression analyses, and a *p*-value of <0.05 was considered statistically significant. All data analyses were conducted using version 9.4 of the SAS statistical package (SAS Institute, Inc., Cary, NC, USA) and version 12.0 of STATA (StataCorp, College Station, TX, USA).

## 3. Results

A total of 5879 women aged 30 and above were included in this study. As shown in Table 2, the average age at enrollment was 49.74 ± 13.91 years; 26.6% lived in Taipei, New Taipei, and Keelung; 30% lived in Taoyuan–Hsinchu–Miaoli/Taichung–Changhua–Nantou; 43.4% lived in other areas; 48.5% had an education level of elementary school and under; 14.8% had an education level of junior high school; and 36.7% had an education level of senior high school and above. The majority of subjects were married (76%), 14% were widowed, and 10% had another relationship status; 25% had a household monthly income of USD 1000 or less, 43% had a monthly household income of USD 1001–2333, and 32% had a monthly household income of USD 2334 or more.

In Table 1, the univariate analysis results show that parity (≥4 children vs. nulliparous, crude HR = 0.41 (95% CI: 0.19–0.89), or per-unit increase crude HR = 0.84 (0.74–0.96), BMI (≥27 kg/m^2^ vs. <24 kg/m^2^; crude HR = 1.77; 95% CI: 1.09–2.88), and time spent sitting (≥3 h/day vs. 0–2.9 h/day; crude HR = 2.04; 95% CI: 1.17–3.57) were significantly associated with female BC. Alcohol intake (crude HR = 0.49; 95% CI: 0.24–1.01) and sugary drink intake (crude HR = 1.45; 95% CI: 0.96–2.19) were marginally associated with female BC. Therefore, we included the above variables and age in the multivariate analysis. When adjusted for multiple comparisons, parity (HR = 0.63; 95% CI: 0.44–0.91), BMI (HR = 1.34; 95% CI: 1.04–1.71), and time spent sitting (≥3 h/day vs. 0–2.9 h/day; HR = 1.89; 95% CI: 1.08–3.32) were statistically significantly associated with female BC (Table 3).

Additionally, we assessed the association between time spent sitting and sugary drink intake combined and female BC incidence, as shown in Table 4. After adjusting for age, parity, and BMI, the risk of female BC was positively associated with sitting for ≥3 h/day and consuming sugary drinks (vs. women who sit for 0–2.9 h/day and do not consume sugary drinks; adjusted HR = 2.51; 95% CI: 1.01–6.23).

We followed up 5879 women aged ≥30 years for 72,453 person-years. The incidence of female BC was 154.6 per 100,000 person-years (95% CI: 127.9–185.3). The median of the follow-up period was 13.25 years. The average age upon diagnosis of female BC was 55.82 ± 10.95 years. During the observation period, the Nelson–Aalen estimate of the cumulative incidence of female BC was 0.1% in one year, 0.3% in three years, 0.8% in five years, and 1.5% in 10 years. The cumulative incidence regarding the time spent sitting ≥3 h/day and sugary drink intake group was significantly higher than that in the time spent sitting 0–2.9 h/day and no sugary drink intake group (log-rank test *p* = 0.027; Figure 2).

## 4. Discussion

This research comprised a population-based cohort study on 5879 Taiwanese women aged 30 years and above. During the 13-year, 72,453 person-year median follow-up, there were 112 new cases of female BC, with an incidence rate of 154.6 per 100,000 person-years (95% CI: 127.9–185.3). The mean age of diagnosis of female BC was 55.82 years, which is similar to the nationwide age of female BC diagnosis (median = 54 years [27]). In our cohort study, the incidence of BC among Taiwanese women was higher than that in a neighboring Asian country, Japan (78 per 100,000 person-years) [28], but lower than that in European and North American countries (339, 232, 355, and 344 per 100,000 person-years in Denmark, the United Kingdom, the United States, and France, respectively) [29,30,31,32]. After adjusting for multiple RFs, the risks associated with female BC were parity (HR = 0.63; 95% CI: 0.44–0.91), BMI (HR = 1.34; 95% CI: 1.04–1.71), and sedentary time (HR = 1.89; 95% CI: 1.08–3.32). Women who sit for ≥3 h/day and consume sugary drinks have a significantly higher risk of developing BC than those who sit for <3 h/day and do not consume sugary drinks (adjusted HR = 2.51; 95% CI: 1.01–6.23). This study has several advantages, including its prospective design, long-term follow-up, and nationally representative study population. This study had a sufficient sample size and follow-up period to evaluate the correlation between female BC and the combination of sedentary time and sugary drink intake. Since this cohort includes only Taiwanese women, some studies showed that there may be ethnic differences in the etiology and biology of BC between Asians and non-Asians [33]. In Taiwan, the younger female BC is characterized by a higher prevalence of luminal A subtype and a lower prevalence of basal-like subtype, unlike in Western countries [34]. Therefore, the results of this study are not applicable to the Western female population.

In our study, sedentary time was the most important predictor of female BC risk. However, there are not many studies that have evaluated the association between sedentary time and the development of female BC. The findings from existing cohort studies have provided little support for such an association. One study found a significantly increased risk of BC with increasing sedentary time (≥12 vs. <5.5 h/day OR = 1.94; 95% CI: 1.01–3.70) in White women [35], whereas another study reported an association between sedentary time and female BC (≥10 vs. <5 h/day HR = 1.27; 95% CI: 1.06–1.53) in African-American women [36]. Nonetheless, a study of American women did not support the association, especially for postmenopausal women (sedentary time ≥10 vs. ≤5 h/day HR = 1.00; 95% CI: 0.92–1.09) [37]. There are three recent studies that associated time spent watching TV with breast cancer. One study indicated that there is a borderline significant (trend *p* = 0.053) in the time spent watching TV to increase the risk of breast cancer in Japanese women [38]. Another study found a lower risk of developing breast cancer in UK women with less time spent watching TV (≤1 vs. 1– ≤ 3 h/day HR = 0.92; 95% CI: 0.85–0.996) [39]. The other study reported an increased risk of breast cancer in Spanish women who watch >2 h/day of TV compared to <1 h (HR = 1.67; 95% CI:1.03–2.72) [40]. Sedentary behavior has been increasing in modern life. In recent years, the effect of sedentary lifestyle on health has drawn public attention [41]. Sedentary behavior may be related to several potential biological mechanisms, such as insulin resistance, obesity, and chronic inflammation, and is considered to play a role in the development of female BC [15,42,43,44]. Sedentary behavior includes prolonged sitting or reclining, which reduces energy expenditure, causing weight gain and obesity [42,45]. It can affect female BC development through adiposity-related pathways [15,44,46]. However, it can also be an independent RF for female BC [43,47]. Sedentary behavior also increases the insulin response to glucose loading [48]. In a bed rest study, it was proven that an increase in sedentary time can cause a series of harmful metabolic effects, such as a significant decrease in systemic insulin sensitivity [49]. Other proposed biological mechanisms for sedentary behavior include the reduction in vitamin D levels [50,51] and an imbalance in sex hormones [52,53].

In a recent French cohort study, it was reported that the consumption of SSB is positively associated with female BC (HR = 1.22; 95% CI: 1.07–1.39) [54]. Two other cohort studies pointed out that the consumption of SSB can increase the postmenopausal risk for women. The incidence of BC in postmenopausal women was 1.21 (95% CI: 1.03–1.43) times more for those who consume between one and six sugar-sweetened soft drinks per week relative to those who consume less than one per month in an Australian cohort study [55]. A Spanish cohort study showed that BC incidence in postmenopausal women who regularly consume sugary drinks is 2.12 (95% CI: 1.02–4.41) times higher than in those who never or rarely consume sugary drinks [56]. One systematic review indicated that each serving per day increase in sugary drinks is associated with a weight gain of 0.22 (95% CI: 0.09–0.34) kg per year. Carbonated beverages, soda, sweetened beverages, fruit drinks, and sports drinks may result in excess energy intake, causing obesity [57]. In addition, one study using a U.S. cohort found that, with adjustments including a change in weight, SSB consumption frequency was classified as none to less than one serving per month, one serving per month to less than one serving per week, one serving per week to one serving per day, and greater than or equal to one serving per day. Visceral adipose tissue (VAT) volume increased by 658, 649, 707, and 852 cm^3^ (95% CI: 602–713, 582–716, 657–757, and 760–943) (trend *p* < 0.001), respectively, indicating that SSBs may lead to an increase in VAT but not weight [58]. One hospital-based case–control study found that a higher visceral adiposity index (VAI ≥1.72) (OR = 1.91; 95% CI: 1.17–3.13) is associated with female BC, indicating that visceral adipose tissue accumulation increases the incidence of female BC [59].

Sedentary behavior and sugary drink intake combined are significantly positively associated with female BC. This study assessed sedentary behavior based on self-reported questionnaires regarding the total time spent sitting in a day, including working, watching TV, and using the computer. This study’s prospective design helped to reduce the effect of recall bias. However, it is limited by not using an objective measurement method to determine time spent sitting, and we did not collect more detailed information about sedentary behavior. A recent study suggested that sedentary time patterns may affect health outcomes more than total time spent sitting [60].

This study adopted a prospective design; therefore, reverse causality bias resulting from pre-existing conditions is not likely to occur, which is one of the main strengths of this study. Moreover, the subjects of our study were recruited from a nationwide sample of Taiwanese residents, enabling us to provide risk assessments and prevention strategies for female BC in Taiwanese. The quality of the NHIRD’s Registry for Catastrophic Illness ensures the accuracy and completeness of our results in identifying incident cancer cases [23,24]. The first limitation in the current study is the small number of female BC cases. The power of the test is slightly insufficient, such as for smoking and exercise, which need to be verified in future studies with large samples. Consequently, the significant results observed in this study may be by chance. Despite that, the number of incident cases was low, the large number of subjects in the reference group allowed for the estimation to have sufficient precision. Second, our results were based on a single measurement of time spent sitting and sugary drink intake and other RFs only at baseline. Because of the limitations of not using an objective measurement method (such as accelerometry) to determine the time spent sitting, we did not collect more detailed information about sedentary behavior. We also did not assess weight gain or lifestyle changes over the follow-up period. Third, given the strong negative correlation between breastfeeding and the risk of female BC, we did not consider the duration of estrogen interruption to calculate lifetime exposure parity for endogenous estrogen. Therefore, the risk associated with estrogen exposure may have been underestimated in our study. Lastly, due to the low number of incident cases, we had to consider female BC cases as a whole, rather than dividing them into histological subtypes for analysis. The potential for heterogeneity in female BC associated with RFs could not be inspected in this study.

## 5. Conclusions

Our findings provide evidence that other RFs, such as parity, BMI, and time spent sitting, may also be implicated in female BC. In contrast, time spent sitting was the dominant predictor of female BC risk in the study cohort. Our data also indicate that time spent sitting combined with sugary drink intake may be a modifiable RF for female BC. Since time spent sitting and sugary drink intake are modifiable factors in managing health, the findings of this study support the recommendation for specific lifestyle modifications to reduce the risk of female BC.

## Figures and Tables

**Figure 1 cancers-14-00788-f001:**
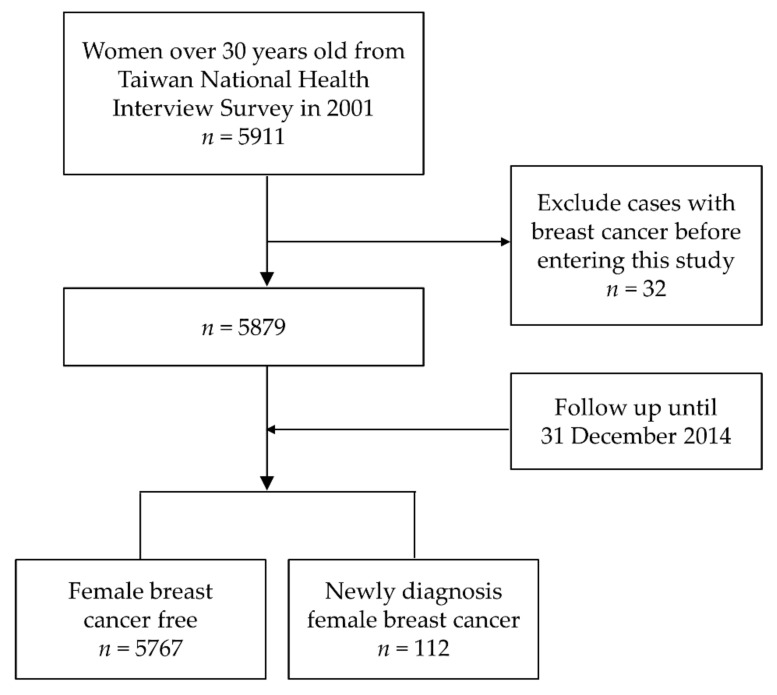
Flow chart for selecting the study participants.

**Figure 2 cancers-14-00788-f002:**
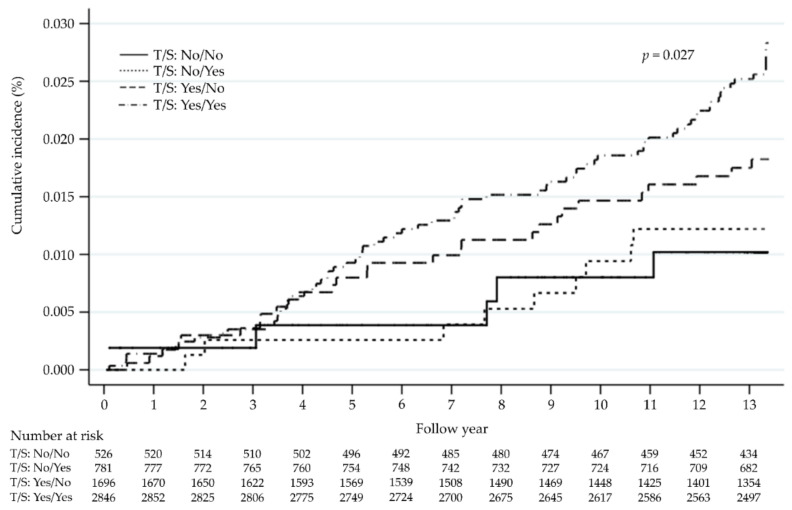
Cumulative incidence of female breast cancer based on time spent sitting (T) and sugary drink intake (S).

**Table 1 cancers-14-00788-t001:** Risk factors associated with the incidence of female breast cancer.

Variables	Items	Cases/Person-Years	95% CI of HR ^1^	HR ^1^ (95% CI)	*p*
Hypertension	No	95	63,428	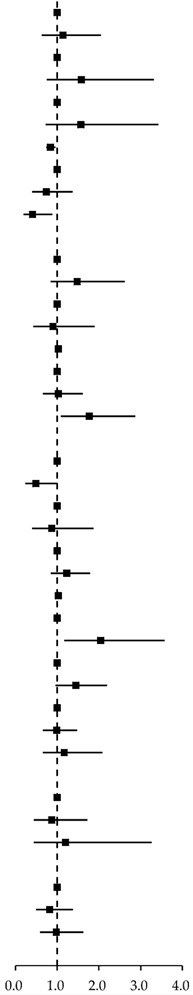	Referent	
	Yes	17	9026	1.14	(0.63–2.05)	0.669
Diabetes	No	104	69,133	Referent	
	Yes	8	3320	1.58	(0.75–3.32)	0.230
Hyperlipidemia	No	105	69,571	Referent	
	Yes	7	2882	1.57	(0.72–3.43)	0.262
Parity	Continuous			0.84	(0.74–0.96)	0.009
	Nulliparous	12	5776	Referent	
	1–3 children	75	45,331	0.74	(0.40–1.37)	0.341
	≥4 children	25	21,346	0.41	(0.19–0.89)	0.024
	*p* for trend			0.013	
Hormone therapy	No	98	66,187	Referent	
	Yes	14	6267	1.48	(0.84–2.62)	0.173
Postmenopause	No	80	52,546	Referent	
	Yes	32	19,907	0.90	(0.43–1.90)	0.784
Body mass index	Continuous			1.03	(0.98–1.08)	0.278
	<24 kg/m^2^	55	40,029	Referent	
	≥24 to <27 kg/m^2^	33	22,679	1.03	(0.66–1.62)	0.882
	≥27 kg/m^2^	24	9745	1.77	(1.09–2.87)	0.020
	*p* for trend			0.044	
Alcohol intake	No	104	62,533	Referent	
	Yes	8	9921	0.49	(0.24–1.01)	0.052
Smoking	No	105	67,225	Referent	
	Yes	7	5228	0.87	(0.40–1.87)	0.717
Exercise	No	50	36,295	Referent	
	Yes	62	36,158	1.23	(0.85–1.79)	0.269
Time spent sitting	Continuous			1.03	(0.98–1.09)	0.229
	0–2.9 h/day	14	16,291	Referent	
	≥3 h/day	98	56,162	2.04	(1.17–3.57)	0.013
Sugary drink intake	No	33	26,726	Referent	
	Yes	79	45,728	1.45	(0.96–2.19)	0.080
Meat intake	Almost every day	43	28,407	Referent	
	1–5 times per week	52	34,693	0.99	(0.66–1.48)	0.941
	<1 time per week	17	9353	1.17	(0.66–2.08)	0.584
	*p* for trend			0.688	
Vegetable intake	Almost every day	99	63,665	Referent	
	3–5 times per week	9	6642	0.87	(0.44–1.73)	0.697
	≤2 times per week	4	2146	1.20	(0.44–3.26)	0.724
	*p* for trend			0.981	
Fruit intake	Almost every day	75	46,777	Referent	
	3–5 times per week	18	13,678	0.82	(0.49–1.38)	0.460
	≤2 times per week	19	11,999	0.98	(0.59–1.63)	0.939
	*p* for trend			0.765	

^1^ Adjusted for age in the Cox proportional hazards regression.

**Table 2 cancers-14-00788-t002:** Demographic characteristics of the women aged ≥30 years included in this study (*n* = 5879).

Variable Group	*n*	(%)
Age (years)		
30–39	1724	(29.3)
40–54	2334	(39.7)
≥55	1821	(31.0)
Continuous (mean ± standard deviation)	49.73 ± 13.91	
Area		
Taipei, New Taipei, Keelung	1561	(26.6)
Taoyuan–Hsinchu–Miaoli, Taichung–Changhua–Nantou	1765	(30.0)
Other	2553	(43.4)
Education		
Elementary school and under	2849	(48.5)
Junior high school	870	(14.8)
Senior high school and above	2160	(36.7)
Marital status		
Married	4464	(75.9)
Widowed	809	(13.8)
Other	606	(10.3)
Household monthly income		
USD ≤1000	1475	(25.1)
USD 1001–2333	2525	(42.9)
USD ≥2334	1879	(32.0)

**Table 3 cancers-14-00788-t003:** Multivariate analysis for female breast cancer incidence.

Variables	HR ^1^ (95% CI)		*p*
Age (years)	1.01	(1.00, 1.03)	0.111
Parity (ordinal)	0.63	(0.44, 0.91)	0.013
Body mass index (ordinal)	1.34	(1.04, 1.71)	0.022
Alcohol intake	0.50	(0.24, 1.02)	0.056
Time spent sitting ≥3 h/day	1.89	(1.08, 3.32)	0.026
Sugary drink intake	1.46	(0.96, 2.21)	0.074

^1^ Cox proportional hazards regression covariates: age, parity (nulliparous, 1–3 children, ≥4 children), body mass index (<24, ≥24 to <27, ≥27 kg/m^2^), alcohol intake (yes/no), time spent sitting ≥3 h/day (yes/no), and sugary drink intake (yes/no).

**Table 4 cancers-14-00788-t004:** Combined effects of time spent sitting and sugary drink intake on the incidence of female breast cancer.

Time Spent Sitting ≥3 h/day	Sugary Drink Intake	HR ^1^ (95% CI)		*p*
No	No	Referent		
No	Yes	1.22	(0.41, 3.64)	0.726
Yes	No	1.69	(0.65, 4.37)	0.282
Yes	Yes	2.51	(1.01, 6.23)	0.048

^1^ Cox proportional hazards regression, covariates: Age, parity (ordinal) (nulliparous, 1–3 children, and ≥4 children), body mass index (ordinal) (<24, ≥24 to <27, and ≥27 kg/m^2^).

## Data Availability

The datasets generated and/or analyzed during the current study are not publicly available in accordance with the policy of the Health and Welfare Data Science Center, Ministry of Health and Welfare, Taiwan, but are available from the corresponding author upon reasonable request.

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
