# Peer review of "Risk Factors for Female Breast Cancer: A Population Cohort Study"

_cancers, 2022, doi:10.3390/cancers14030788_

Round 1
Reviewer 1 Report
This study by Wang and coworkers aims at identifying risk factors of female breast cancer. This study is based upon a Taiwanese cohort of > 5800 women over 30 years old. Among parameters taken into account in this study, authors have mainly focused on sedentary lifestyle and drinking sugar-sweetened beverages (SSB).
This work is of interest. However several points need to be clarified, assertions modified and conclusions revisited.
Major points.
1) Authors do not define what "sedentary lifestyle" means and includes in terms of behavior. This should be introduced thoroughly, as soon as the introduction section.
2) p 2 : "It negatively affects the quality of life of patients and their families" (about BC). ANY cancer of ANY organ would affect the quality of life of patients and their families. Please amend.
3) p 2 : "There are many causes of female BC" : already specify here that only modifiable RFs are taken into account. Otherwise you need to report about non modifiable RFs : higher breast density, BRCA mutations, family history of breast cancer...
4) Ref #8 must be considered only valid for Japanese women, please do not make it general (e.g. Rosner & Colditz Ann Epidemiol 2021)
5) "Drinking an extra 10g alcohol"... : what does "extra 10g alcohol" mean ? "Extra" on baseline alcohol consumption ??? Please clarify.
6) p 4 : "A two-tailed test"... which test ? T test ? Please clarify.
7) Table 2 : the legend is not informative enough. On the scale between 0.0 and 3.6, factors considered as protective against cancer in general (e.g. fruit intake 3-5 times a week) fall on the same side as smoking. Conversely, practising exercise falls on the same side as SSB intake and time spent sitting risks. Please explain, and describe more thoroughly the statistical analyses performed here, since not every (average) reader is familiar with this type of statistical tests and their analyses.
7) Discussion : authors should start with 2 points :
- the fact that their cohort only included Taiwanese women, who could have a peculiar genetic background (like Japanese cited in ref #8) that gives results not applicable to the general population;
- the fact that NO objective measurement method was used to assess time spent sitting (data are then only based upon declarative assertions from the women included in this cohort), and that NO detailed information about sedentary behavior had been collected. Again, at this stage, authors should explicit what "sedentary lifestyle" corresponds to, specially since practising exercise falls on the same side as SSB intake in Table 2.
Reviewer 2 Report
The manuscript ID: cancers-1539130 entitled “Risk Factors for Female Breast Cancer: A Population Cohort Study” is focused on the association of a sedentary lifestyle and consumption of sweet beverages and the breast cancer risk in Taiwanese women. The authors should better discuss potential mechanisms underlying the association between these risk factors and breast cancer.
References are old; the most recent (two) are of 2020. I would appreciate a further deepening. The existence of an association between sedentary time and the female BC onset is sustained by the references 32 and 33. However, there is another paper of Nomura et al. published in 2017 that not support an association between sedentary time and breast cancer incidence.
To make their study more convincing, the authors should provide readers with more information about, for example, the questionnaire that was administered to the study participants. It will sufficient to insert the questionnaire as a figure.
In Materials and Methods section, authors (2.1 Study Cohort and Baseline Characteristics) say that 5911 women from the largest 2001 NHIS cohort were enrolled. I found the Figure 1 a bit confusing; I think that it would be better to initiate the flow chart for selecting the study participants from the box of women over 30 years old.
In general the manuscript is well-written; please check all abbreviations.
